# PADriver: Towards Personalized Autonomous Driving

## Abstract

In this paper, we propose PADriver, a novel closed-loop framework for personalized autonomous driving (PAD). Built upon Multi-modal Large Language Model (MLLM), PADriver takes streaming frames and personalized textual prompts as inputs. It autoaggressively performs scene understanding, danger level estimation and action decision. The predicted danger level reflects the risk of the potential action and provides an explicit reference for the final action, which corresponds to the preset personalized prompt. Moreover, we construct a closed-loop benchmark named PAD-Highway based on Highway-Env simulator to comprehensively evaluate the decision performance under traffic rules. The dataset contains 250 hours videos with high-quality annotation to facilitate the development of PAD behavior analysis. Experimental results on the constructed benchmark show that PADriver outperforms state-of-the-art approaches on different evaluation metrics, and enables various driving modes.

## 1 Introduction

In real-world driving, users may prefer to choose different autonomous driving styles that are determined based on user preferences (Jin et al., 2023a; Cui et al., 2024; Sha et al., 2023; Chen et al., 2023). For example, passengers in a hurry may opt for a higher speed to reach their destination, while others may prefer a slower yet more comfortable journey. Therefore, personalized autonomous driving (PAD) is crucial, which aims to adjust the autonomous driving mode based on the drivers' preferences. PAD can be assessed based on two key elements: *speed* and *comfort*. Speed is mainly related to the driver's character, and is also limited by the traffic rules. Frequent lane changes, acceleration, and deceleration affect the driving experience of different drivers - comfort. Previous works for personalized driving (Ling et al., 2021; Schrum et al., 2024) are oriented towards individual users. They face challenges in generalizing the needs of different human groups.

Most existing end-to-end driving models (Chitta et al., 2021; Prakash et al., 2021; Wu et al., 2022; Hu et al., 2023) are generally trained within a single driving mode by mimicking a single expert. However, PAD needs to be satisfied with different personalized modes, which makes it difficult for these methods to learn multiple distributions in the same model. Recently, knowledge-driven large language models (LLMs) (OpenAI, 2023b;a) show a set of capabilities for task reasoning, state prediction, and action planning based on textual detection results (Fu et al., 2023; Wen et al., 2023; Jin et al., 2023a; Cui et al., 2024; Sha et al., 2023; Jin et al., 2023b; Chen et al., 2023), which shows the potential for personalized planning with different modes. However, sparse textual representation may make it difficult to fully provide enough scene information for decision and planning tasks. Therefore, the Multi-modal Large Language Models (MLLMs) using the text and visual information (Radford et al., 2021; Li et al., 2023; Alayrac et al., 2022; Liu et al., 2023b;a; OpenAI, 2023c), are necessary to enable comprehensive scene understanding. However, those MLLM-based studies (Wang et al., 2023a; Shao et al., 2023; Wang et al., 2023b) only make some short-term action adjustments according to human notifications, without involving the driving modes switching.

To provide users with personalized driving experiences, we introduce PADriver, a personalized closed-loop autonomous driving (AD) framework based on MLLMs. The PADriver is designed with three modes: *slow*, *normal*, and *fast*. Users can directly change the mode by setting different personalized prompts. To make an appropriate decision based on the surrounding environment, we introduce the concept of *danger level* that assigns a danger score to each potential action. PADriver takes the proper

action associated with the output scene description, personalized prompt, and estimated danger level. To ensure smooth planning, we also introduce the ego states queue that stores the ego states over the past frames, serving as part of textual prompts. Moreover, we establish a new benchmark based on the Highway-Env simulator (Leurent, 2018) to comprehensively evaluate the performance of different closed-loop methods, which can avoid the expensive cost of real-world data collection and safety evaluation. The benchmark PAD-Highway contains 250 hours data with high-quality annotation to encourage the development of personalized driving.

Our PADriver has the following advantages compared to existing approaches: (1) Through the personalized prompt, PADriver can perform different driving modes at any time, given the preference of users. (2) PADriver serves as the first work to explicitly model the danger level of the corresponding action among all existing MLLM-based methods. The estimated danger level provides an important reference for the final decision, associated with driving mode.

In summary, our contributions are listed as follows:

- We present a novel closed-loop driving system for personalized driving. The danger level is further introduced to model the risk of potential actions.

- A new benchmark PAD-Highway containing dataset and metrics is introduced to comprehensively evaluate the performance of different closed-loop methods.

- Our approach with slow mode achieves state-of-the-art performance, while other modes provide some potential choices for users.

## 1.1 RELATED WORKS

### 1.1.1 END-TO-END AUTONOMOUS DRIVING MODELS

The evolution of autonomous driving technology is characterized by the development of data-driven approaches, especially for the end-to-end models (Chitta et al., 2021; Prakash et al., 2021; Wu et al., 2022), which simplify the overall process by directly translating sensory inputs into control commands.

Some recent advances further propel the end-to-end driving by enhancing perception, improving occlusion detection, and integrating multi-modal sensor information. ST-P3 (Hu et al., 2022) aligns temporal BEV features to construct a dense cost map, utilizing hand-crafted rules to derive the optimal planning trajectory. UniAD (Hu et al., 2023) integrates multiple scene prediction modules by employing task queries as the interfaces, which facilitate the connection of each prediction module with a transformer decoder structure, leading to the final planning stage. VAD (Jiang et al., 2023) employs a strategy to vectorize and regularize the planning trajectory alongside the driving scene, thereby effectively incorporating vectorized scene information to enhance the planning process. ThinkTwice (Jia et al., 2023b) uses a scalable decoder, incorporating the dense and spatial-temporal priors to extract information from critical regions. This approach leverages the identified features to improve the precision of coarse action predictions through meticulous refinement.

### 1.1.2 MULTI-MODAL LARGE LANGUAGE MODELS

The advent of Large Language Models (LLMs), including Palm (Chowdhery et al., 2023), Vicuna (Chiang et al., 2023; Peng et al., 2023), LLaMA (Touvron et al., 2023a;b), and GPTs (Radford et al., 2018; 2019; OpenAI, 2023a;b), marks a pivotal shift in artificial intelligence research. Simultaneously, the emergence of Multi-modal Large Language Models (MLLMs) like CLIP (Radford et al., 2021), BLIP-2 (Li et al., 2023), Flamingo (Alayrac et al., 2022), the LLaVA series (Liu et al., 2023b;a), and GPT-4V (OpenAI, 2023c) represents a substantial advancement in the field. These models have extended the boundaries of traditional LLMs by incorporating sophisticated reasoning abilities with the capability to process and interpret images, point clouds, and other modalities. Furthermore, the LLaVA series (Liu et al., 2023b;a) introduce a visual instruction tuning method, which has demonstrated superior performance in tasks requiring perception and spatial reasoning. These models, trained on extensive and diverse datasets derived from the webs, have successfully injected the common-sense knowledge applicable to a wide range of domains, such as robotics (Brohan et al., 2023) and autonomous driving.

### 1.1.3 MLLM FOR INTELLIGENT VEHICLES

Recent researches (Jin et al., 2023a; Xu et al., 2023; Chen et al., 2023; Jin et al., 2023b; Cui et al., 2024) have integrated intelligent agent with LLMs to enhance real-world performance, leveraging spatial reasoning and perception capabilities. For instance, Talk2BEV (Dewangan et al., 2023) introduces an innovative Bird's-Eye View (BEV) representation, merging visual and semantic information to facilitate Visual Question Answering (Visual QA), which encompasses spatial and visual reasoning tasks. Similarly, LiDAR-LLM (Yang et al., 2023) employs LLMs to process point clouds, addressing 3D captioning, grounding, and QA tasks.

MLLMs are further extended to explore the planning and decision tasks (Liu et al., 2023c; Mao et al., 2023; Sha et al., 2023). Drive-like-Human (Fu et al., 2023) reconsiders the constraints of existing autonomous driving frameworks and proposes a paradigm shift through the integration of diverse LLM APIs. Dilu (Wen et al., 2023) further combines reasoning and reflection modules to make informed decisions based on common-sense knowledge and the accumulated experience of GPTs. However, these methods utilize LLMs and rule-based priors to translate scenes into text, demonstrating the scene understanding and logical reasoning capability of LLMs. LMDrive (Shao et al., 2023) aims to integrate MLLMs with multi-modal sensor data, enabling the system to make decisions based on a comprehensive understanding of both environment and natural language guidance. It also integrates the alerts from human with navigation system. DriveMLM (Wang et al., 2023a) introduces a closed-loop MLLM-based system for behavior planning in Carla simulators (Dosovitskiy et al., 2017). It merges the driving regulations, user commands, and sensor inputs to guide the driving decisions and provide explanations.

Some previous approaches (Cui et al., 2024; 2023) based on LLMs are related with the concept of PAD. Driven by the human commands, they frequently adjust the temporary actions. Though their efforts show the potential of MLLMs or LLMs for personalized driving, our work has significant differences with them. These methods primarily focus on addressing the preferences of specific users, posing challenges to widespread application. In contrast, our PADriver integrates multiple driving modes within a single MLLM-based framework to accommodate personalized and diverse driving preferences for general scenes.

## 2 METHODOLOGY

In this section, we introduce PADriver, a novel framework designed for personalized autonomous driving, enabling closed-loop control of autonomous vehicles. First, we show the overall architecture of our framework. Next, the textual prompts, which are input to the PADriver system, are presented in detail. Lastly, we describe how the final action is made based on the danger level estimation.

### 2.1 OVERALL ARCHITECTURE

As illustrated in Figure 1, our PADriver, is built upon the multi-modal large language models (MLLMs) and evaluated on the closed-loop simulator Highway-Env. Given time step $t$, the current Bird's Eye View (BEV) frame $I_t$ and textual prompts are input to the PADriver system. The BEV frame is tokenized into vision tokens by the vision encoder (e.g., CLIP (Radford et al., 2021)).

The vision tokens are then projected into the word embedding space for alignment through a linear layer. Simultaneously, the textual prompts, including the system prompt <SYSTEM>, personalized prompt <PERSONALIZE>, and ego state prompt <EGO_STATE>, are tokenized by the large language models(LLMs) tokenizer (e.g., LLaMA (Touvron et al., 2023a;b)), which converts natural language prompts into textual tokens. Vision tokens and textual tokens are concatenated together and further input to the LLM. The LLM autoregressively generates scene descriptions <DESCRIPTION>, danger level assessments <DANGER_LEVEL> for each possible action, and the final action decision <ACTION>. The final action is implicitly associated with both danger level estimation and scene description. The overall process is illustrated in Figure 2.

### 2.2 TEXTUAL PROMPTS

The overall textual prompts are constructed as follows:

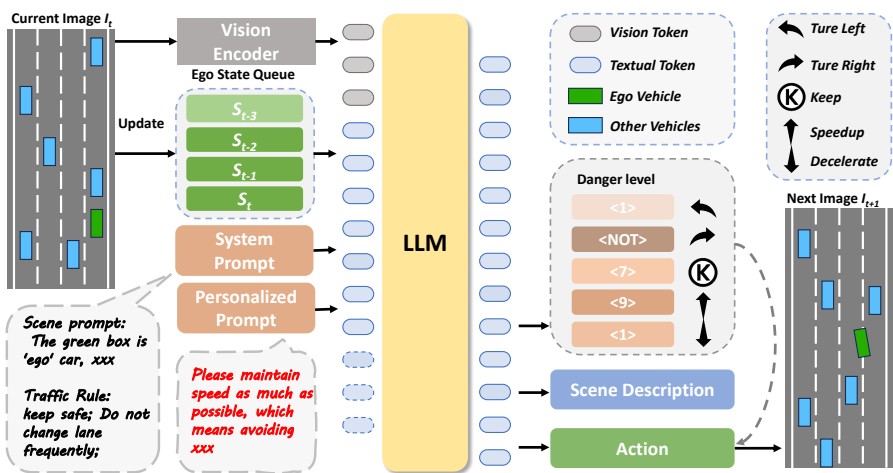

Figure 1: The PADriver framework takes textual prompts(system prompts, personalized prompts and ego state queue) and the BEV image as input, tokenizing them into textual and vision tokens. Based on the multi-modal tokens, LLM then autoregressively generates scene descriptions, assesses the danger level for each potential action, and makes the final action decision. The final action is implicitly affected by both the danger level estimation and the scene description.

$$\text{Prompt}: \text{<SYSTEM>} \text{<PERSONALIZE>} \text{<EGO\_STATE>},$$

where the system prompt <SYSTEM> provides the primary description of the environment, personalized prompt <PERSONALIZE> indicates the driving mode, and ego state prompt <EGO_STATE> provide the ego historical information includes the speed and coordinates of the ego-car.

**System Prompts:**

The system prompts <SYSTEM> mainly include three components: 1) the basic environment descriptions of the ego car and other cars, 2) the traffic rules to inform the model of actionable constraints, which indicate when to take or avoid the corresponding action, and 3) other appropriate notices and instructions that may help the system.

**Personalized Prompts:** The personalized prompts <PERSONALIZE>, designed to reflect the agent's driving preferences, are categorized into three distinct modes: *slow*, *normal*, and *fast*. The fast mode is designed to prioritize speed while ensuring safety. Conversely, the slow mode represents the standard setting, focusing on minimizing operation frequency. The normal mode is the most comfortable one, achieving a good trade-off between the speed and safety.

**Ego State Prompts:** The ego state prompts <EGO_STATE> constitute a critical element of the PADriver. Unlike prior works (Jia et al., 2023a) that rely on image-action pair logs, our strategy utilizes a memory queue of ego states. The queue includes $n$ historical states from preceding frames, encompassing the velocity $v$ and the coordinates of the ego vehicle $(x, y)$. The ego state queue can be represented as $\{(v_t, x_t, y_t), (v_{t-1}, x_{t-1}, y_{t-1}), \ldots, (v_{t-n}, x_{t-n}, y_{t-n})\}$. The ego state queue is updated at each time-step and is readily accessible through the Inertial Measurement Unit (IMU) (Faisal et al., 2019) of the ego vehicle.

## 2.3 System Outputs

The MLLM outputs include scene description <DESCRIPTION>, danger level estimation <DANGER_LEVEL> for possible actions and final action decision <ACTION>.

The outputs can be structured as follows

$$\text{Answer}: \text{<DESCRIPTION>} \text{<DANGER\_LEVEL>} \text{<ACTION>} \text{<STOP>}$$

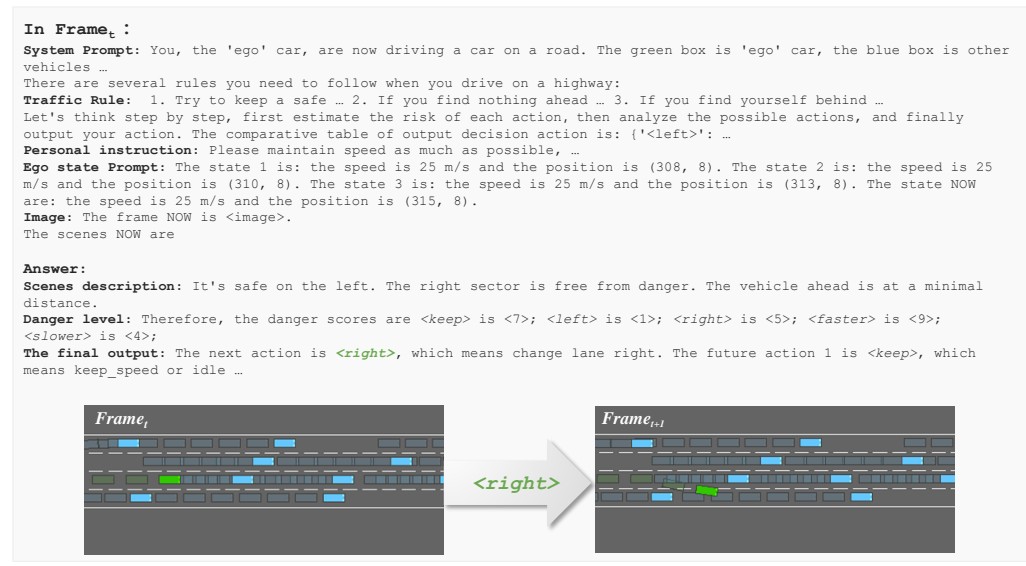

```
In Frame_t :
System Prompt: You, the 'ego' car, are now driving a car on a road. The green box is 'ego' car, the blue box is other
vehicles …
There are several rules you need to follow when you drive on a highway:
Traffic Rule:  1. Try to keep a safe … 2. If you find nothing ahead … 3. If you find yourself behind …
Let's think step by step, first estimate the risk of each action, then analyze the possible actions, and finally
output your action. The comparative table of output decision action is: {'<left>': …
Personal instruction: Please maintain speed as much as possible, …
Ego state Prompt: The state 1 is: the speed is 25 m/s and the position is (308, 8). The state 2 is: the speed is 25
m/s and the position is (310, 8). The state 3 is: the speed is 25 m/s and the position is (313, 8). The state NOW
are: the speed is 25 m/s and the position is (315, 8).
Image: The frame NOW is <image>.
The scenes NOW are

Answer:
Scenes description: It's safe on the left. The right sector is free from danger. The vehicle ahead is at a minimal
distance.
Danger level: Therefore, the danger scores are <keep> is <7>; <left> is <1>; <right> is <5>; <faster> is <9>;
<slower> is <4>;
The final output: The next action is <right>, which means change lane right. The future action 1 is <keep>, which
means keep_speed or idle …
```

Figure 2: The textual prompts and step-by-step reasoning output are generated by the MLLM in Frame $T$. The textual input includes the system prompts (traffic rules and basic environmental descriptions), personalized prompts, and ego state prompts. The system outputs the scene description, danger level estimation, and the final action output. The model generates the final action `<right>` in fast mode in frame $t$. The response from the environment occurs in frame $t + 1$.

where `<STOP>` denotes the termination token. The output from the MLLMs follows the theory of the chain of thought (CoT) (Wei et al., 2022). The step-by-step thinking rather than directly outputting the final action, alleviates the hallucination phenomena of LLM and improves logical reasoning (Wen et al., 2023).

**Scene Description:** The scene description includes some concise key information, such as the presence of adjacent vehicles parallel to the ego vehicle and the situation of the vehicle ahead. The scene description provides the reference for the estimation of danger level, which is described in detail next.

**Danger Level Estimation:** For the decision-making process, humans tend to evaluate the risk of various actions. In analogy to human perception, we consider introducing the concept of danger level for MLLM. The MLLM is trained to assign a danger level to each potential driving action based on the scene description. Danger levels are explicitly modeled based on a comprehensive analysis of the Bird's Eye View (BEV) scenes. It incorporates some factors, such as the anticipated reaction time, the velocity of the ego vehicle, and the surrounding environmental conditions—namely, the proximity to and velocity of nearby vehicles. The detailed algorithm for danger level estimation is provided in the Appendix.

Following common practices, the actions include `<left>`, `<keep>`, `<right>`, `<faster>`, and `<slower>`. Each potential action is assigned with a danger level $D \in \mathbb{R}^1$, belonging to the level set {`<0>`, `<1>`, ..., `<8>`, `<9>`, `<NOT>`}. The degree of danger is increased with the increasing of level number. Notably, `<NOT>` indicates an action that is not viable due to safety considerations.

For example, as shown in Figure 2, turning left is prohibited when an adjacent vehicle is present, to prevent collisions. Therefore, the output is: `<keep>` is `<7>`; `<left>` is `<1>`; `<right>` is `<5>`; `<faster>` is `<9>`; `<slower>` is `<4>`.

**Final Action:** We predict the action sequence $A_i \in A \in \mathbb{R}^{m+1}$ for the current frame and future $m$ frames. During training, we use the ground-truth actions of $m + 1$ frames to supervise the action outputs. The dense supervision improves the stability and safety in long-term planning. While during inference, only the predicted action of current frame `<ACTION>` $\in \mathbb{R}^1$ is utilized for decision-making.

Notably, the final action is determined by the personalized prompt based on the danger level. For example, In the fast mode, as shown in Figure 2, when the predicted danger level set is `<keep>` is `<7>`; `<left>` is `<1>`; `<right>` is `<5>`; `<faster>` is `<9>`; `<slower>` is `<4>`, we can take the action of `<keep>` to take more risk yet safe action.

## 2.4 MODEL TRAINING

Like the training schedule of previous MLLMs, our PADriver follows two-stage training process: Pretraining and Supervised Fine-Tuning (SFT). The capability of MLLMs usually benefits from the pretraining on a large amount of data. Such large-scale pretraining enables MLLMs to acquire a general understanding of the ego vehicle's surroundings as well as the risk estimation of different actions. Additionally, the SFT process is designed to refine the MLLM's ability to comprehend the wide array of human decision-making knowledge in driving scenarios. MLLMs can be trained to imitate the personalized behaviors using a small amount of SFT data. Therefore, we first pretrain our framework with large-scale data and then apply SFT with a small amount of data. Details are provided in Section 4.1.

## 3 PAD-HIGHWAY BENCHMARK

We introduce a comprehensive benchmark to conduct fair comparison. The benchmark is built based on the Highway-Env simulator (Leurent, 2018) for closed-loop evaluation. First, we define the detailed settings for the closed-loop evaluation, ensuring reproducibility. Subsequently, we introduce a giant driving decision dataset PAD-Highway to encourage the development of PAD analysis. Finally, we present a set of evaluation metrics designed to thoroughly assess performance from multiple perspectives, including efficiency, safety, and comfort.

### 3.1 BASIC SETTING

The simulated scene is generated with a duration of 30 seconds and a frequency of 10 Hz. The experiment is deployed within a simulated environment featuring a four-lane motorway. Each scene totally has 30 vehicles with a vehicle density of 2.0, and the maximum speed of each vehicle is 30 m/s for regulatory issues of traffic rules. Except for the aforementioned specifications, all experimental conditions follow the default configurations provided by the Highway-Env simulator.

The Highway-Env simulator can theoretically generate an infinite number of scenes given the seeds. For evaluation, we select the foremost 30 continuous seed numbers, ranging from 0 to 30, differing from Dilu (Wen et al., 2023) that randomly selects 10 seeds from a manually predetermined set. The remaining seeds are used to generate the scenes for training data collection.

### 3.2 PAD-HIGHWAY DATASET

We introduce a comprehensive data generation pipeline to construct PAD-Highway dataset. We utilize the Highway-Env simulator to collect Bird's Eye View (BEV) frames, as well as the states, which include the action, speed, and coordinates of both the ego and other vehicles for each frame. Our methodology is divided into two principal collection modes: the *rule-based* and *human-based* collection modes.

***Rule-based* Part:** This collection accelerates the process of accumulating large amounts of data through either random or rule-based actions. The rule is set based on the current states (coordinates and speed) of all cars. For example, if the ego car is too close to the front car, the first priority is to change the lane to avoid a crash when the side lane is safe. The second priority is to slow down. Moreover, the past actions of ego car are also considered, to avoid frequent lane and speed changes.

***Human-based* Part:** The *human-based* collection mode focuses on acquiring data reflective of human driving behaviors to enhance the LLM's comprehension of diverse human decision-making strategies in driving. We develop an annotation system for human-based collection and arrange nearly 20 people to drive the ego car using the Highway-Env simulator, to collect the decisions of human driving behaviors. During the collection stage, each person first finishes the 30s driving and then

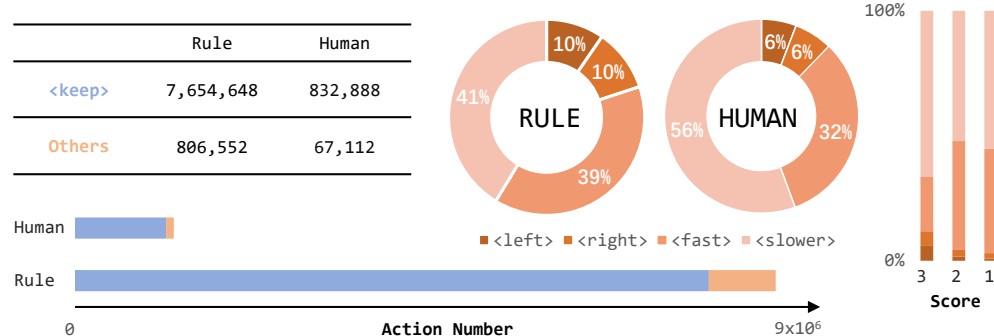

Figure 3: The distribution of the training dataset, which includes the *rule-based* set and *human-based* set, is presented with five actions. From left to right, the first visualization shows the distribution of <KEEP> and other actions using a clustered bar chart. Then, the distribution of non-<KEEP> actions in each set is illustrated in the middle doughnut chart. Lastly, the non-<KEEP> actions of the human-based set are analyzed based on the percentage of three different scores in the right stacked bar chart.

score this driving experience with three modes: 1) I just follow the car, 2) I occasionally take some surpass actions, 3) I want to get ahead of all other cars.

**Data distribution:** The collected dataset includes 235 hours of data from the rule-based mode and an additional 25 hours from human-based driving. The datasets consist of 32,000 videos, each capturing a continuous 30-second duration of driving scenes, sampled at a frequency of 10 Hz. For every frame, we collect the corresponding actions, along with the BEV image and the state parameters (velocity and coordinates) of each vehicle. All collected data are annotated with key information for scene description. The distribution of the dataset is shown in Figure 3.

Notably, PADriver benefits from pretraining on a large number of *rule-based* collected data, with comprising approximately 28,000 clips (300 frames per clip). Such large-scale *rule-based* collected data is used for PADriver pretraining, and The *human-based* data is used for SFT process.

## 3.3 EVALUATION METRICS

To quantitatively evaluate the performance of our framework, we employ distinct evaluation metrics tailored to the framework under consideration. Different from previous evaluation metrics, we propose multiple metrics from different perspectives, such as average driving distance, speed, success completion number, average vehicle density, safe distance keeping rate, and comfort metrics, to comprehensively evaluate the performance.

**Success Completion Number(Suc.):** This metric quantifies the number of successful completion of a 30-second duration with 30 different seeds, illustrating the overall completions rate.

**Average Driving Distance(Dis.)** and **Speed(Spe.):** *Dis.* is calculated by the average driving distance of all successful cases: $Dis. = \frac{1}{m} \sum_{k=0}^{m} D_k$, where $m$ is the number of successful videos.

The calculation of *Spe.* shares a similar case and can be expressed by: $Spe. = \frac{1}{m} \frac{1}{n} \sum_{k=0}^{m} \sum_{i=0}^{n} S_i^k$, where $S_i^k$ is the speed of the ego car for each frame, measuring the average driving speed corresponding to each frame. Notably, the speed is converted from meters per second (m/s) to kilometers per hour (km/h) to align with real-world vehicle speed representation.

These two metrics facilitate a comprehensive evaluation, surpassing previous metrics like *Success Steps* (Wen et al., 2023), which may not fully verify the effectiveness since adopting continuous deceleration strategies can achieve good performance.

Table 1: Comparative analysis of strategies for different driving modes. The average evaluation for *Average Driving Distance (Dis.)*(m), *Average Driving Speed (Spe.)*(km/h), *Safe Distance Keeping Rate (Saf.)*, *Lane Keep Rate (Kep.)*, *Average Vehicle Density (Den.)*, *Success Number (Suc.)* of 30 seeds in different modes, and runtime per frame(s) (*Time*).

| Evaluation | Dis. | Spe. | Saf. | Kep. | Den. | Suc. | Time |
|---|---|---|---|---|---|---|---|
| DwLLMs (Chen et al., 2023)(ICRA 2024) | 411 | 49.42 | 0.49 | 0.56 | 0.35 | 20 | 48.0 |
| GPT-Driver (Mao et al., 2023)(arxiv 2023) | 428 | 51.40 | 0.45 | 0.52 | 0.48 | 23 | 36.2 |
| Dilu (Wen et al., 2023)(ICLR 2024) | 386 | 46.29 | 0.60 | 0.72 | 0.27 | 28 | 24.1 |
| Slow | **554** | **66.46** | **0.90** | **0.87** | 0.68 | **29** | 1.4 |
| Normal | 603 | 72.47 | 0.91 | 0.92 | 0.89 | 25 | 1.4 |
| Fast | 723 | 86.83 | 0.67 | 0.76 | 1.01 | 17 | 1.4 |

Table 2: Comparative analysis of strategies for different driving modes. The comfort evaluation of the average acceleration in the X and Y directions $(\bar{a}_x, \bar{a}_y)(m/s^2)$, and the jerk in the X and Y directions $(\bar{J}_x, \bar{J}_y)(m/s^3)$ of 30 seeds in different modes.

| Evaluation | Spe. | $\bar{a}_x$ | $\bar{J}_x(e^{-2})$ | $\bar{a}_y$ | $\bar{J}_y(e^{-2})$ |
|---|---|---|---|---|---|
| DwLLMs (Chen et al., 2023)(ICRA 2024) | 49.42 | -0.07 | -0.12 | +0.04 | -0.06 |
| GPT-Driver (Mao et al., 2023)(arxiv 2023) | 51.40 | -0.07 | +0.11 | -0.03 | -0.05 |
| Dilu (Wen et al., 2023)(ICLR 2024) | 46.29 | -0.05 | +0.09 | -0.03 | -0.04 |
| Slow | 66.46 | -0.02 | -0.08 | 0 | -0.03 |
| Normal | 72.47 | **-0.01** | **-0.07** | **0** | **0** |
| Fast | **86.83** | **+0.01** | -0.08 | -0.02 | +0.06 |

**Average Vehicle Density(Den.):** This metric is reflected by the number of cars within a distance interval relative to the ego car. *Den.* is proposed to assess the complexity of the driving environment. It highlights the impact of surrounding vehicle density on driving safety and the likelihood of collisions.

**Safe Distance Keeping Rate (Saf.):** Safety is always the most critical metric in the field of autonomous driving. Therefore, we introduce the *Saf.*, calculated by $\frac{1}{m}\frac{1}{n}\sum\sum(dis_t < dis_s)$, where $dis_t$ denotes the distance to the preceding vehicle and $dis_s$ represents the predefined safe distance threshold. We set the $dis_s = 5m$, which is approximately the length of the ego car. This metric indirectly measures the potential for collisions or accidents.

**Lane Keeping Rate (Kep.):** To evaluate the stability of driving decisions, we simply define the *Kep.* as $\frac{1}{m}\frac{1}{n}\sum(a_T = \text{<KEEP>})$, where $a_T$ represents the set of actions undertaken by the vehicle. This metric evaluates the frequency of stationary actions.

**Comfort Metrics:** To evaluate the comfort level, we introduce some metrics for comprehensively evaluating driving comfort: average acceleration $\bar{a}_x = \frac{1}{m}\frac{1}{n}\sum\sum\frac{v_x(t)-v_x(t-1)}{\Delta t}$ and average jerk $\bar{J}_x = \frac{1}{m}\frac{1}{n}\sum\sum\frac{a_x(t)-a_x(t-1)}{\Delta t}$ in the X direction, with corresponding metrics in the Y direction ($\bar{a}_y$ and $\bar{J}_y$). These metrics aim to further quantify the variability in vehicle dynamics, where consistent speed and minimal abrupt changes are synonymous with comfort levels.

These metrics above provide an evaluation framework for autonomous driving from multiple perspectives, emphasizing not only the efficiency of driving but also the critical aspects of safety and passenger comfort.

## 4 EXPERIMENTS

### 4.1 IMPLEMENTATION DETAILS

We adopt Vicuna-7B-1.5 (Chiang et al., 2023; Peng et al., 2023) as the LLM, which is fine-tuned on LLaMA2 (Touvron et al., 2023b). The CLIP-ViT-Large (Radford et al., 2021), pretrained on a large

Table 3: (a). Comparative analysis of model performance across diverse action prediction lengths in the output. (b). Ablation study on the effectiveness of the scenes description (Sce.) and danger levels (Dan.) of the sates prompts. (c). Ablation study on the accuracy ($Acc.$) and average of danger level ($\bar{D}$) element with each action for taken in the human set of the dataset and the three different mode(fast, normal and slow).

| len. | Spe. | Suc. | Dan. | Sce. | Dis. | Spe. | Den. | Suc. | Mode | Spe. | $\bar{D}$ |
|------|------|------|------|------|------|-------|------|------|--------|-------|------|
| 1 | 93.24 | 10 | | | 573 | 68.76 | 0.76 | 3 | Human | 73.11 | 2.02 |
| 3 | 93.02 | 11 | ✓ | | 660 | 79.38 | 0.93 | 7 | Slow | 66.46 | 1.36 |
| 5 | 85.14 | 15 | | ✓ | 608 | 72.97 | 0.78 | 9 | Normal | 72.47 | 1.75 |
| 10 | 86.83 | **17** | ✓ | ✓ | **723** | **86.83** | 1.01 | **17** | Fast | 86.83 | **3.22** |
| | (a) | | | | | (b) | | | | (c) | |

number of image-text pairs, is utilized as the visual encoder. Two multi-layer perceptrons (MLPs) layers, pretrained by LLaVA-1.5-7B (Liu et al., 2023a), are employed as the visual adapter to align the visual tokens with textual tokens. The input image is of size $336 \times 336$. For the training stage, the length of predicted action sequence $m$ is set to 10. All the experiments are conducted on 8 A100 GPUs. The cross-entropy loss is used for supervision following LLaVA (Liu et al., 2023a).

The pretraining process is conducted for 1 epoch on *rule-based* data with approximately 28,000 clips (300 frames per clip). The Supervised Fine-Tuning (SFT) is performed for 2 epochs on the *human-based* data with approximately 3,000 clips. Both the pretraining and SFT processes adopt the same learning rate $2 \times 10^{-5}$ with the AdamW optimizer. We evaluate our method using the 30 seeds in the benchmark(see Section 3). All ablation studies are evaluated with the fast mode.

## 4.2 THE RESULTS IN CLOSED-LOOP DRIVING

**Personality Dividing** The evaluation of our method is performed with three modes: slow, normal, and fast. Each metric is detailed analysed according to the evaluation metrics specified in Section 3.3. Table 1 reveals that the normal mode achieve better performance on *Saf.* and *Kep.*, indicating higher comfort and safety. Conversely, the fast mode exhibits a significant increase in *Dis.* and *Spe.* at the expense of *Suc.*, showing a trade-off. The fast mode, characterized by higher speeds and driving density, shows an increasing of crash risk. We also compare our method with Dilu (Wen et al., 2023) on our benchmark. Our slow mode outperforms it on all metrics, and we consider Dilu to be a relatively cautious approach. The efficiency comparison is also shown in Table 1, with the runtime per frame (s). Our model achieves the highest efficiency compared to the LLM-based methods.

**Comfort** Table 2 presents the comparison of comfort metrics across different driving modes (slow, normal, fast). Dilu (Wen et al., 2023) shows the larger deceleration in the X direction, indicating that the vehicles experience stronger deceleration, likely due to a more conservative driving style with the action of `<slower>`. In contrast, our slow and normal modes have similar accelerations in the X direction, suggesting a comfortable driving experience. The Fast mode shows positive acceleration in the X direction, indicating a tendency to accelerate, which is consistent with the definition of the fast mode. Across all modes, jerk values in both X and Y directions are relatively low, indicating smooth changes in acceleration without sudden starts or stops, which is beneficial for enhancing comfort. Dilu has a positive jerk in the X direction, whereas our modes have near-zero values, suggesting slightly more abrupt changes in acceleration.

## 4.3 ABLATION STUDY

**Length of Predicted Action Sequence:** We evaluated the impact of varying the sequence length of actions. As indicated in Table 3a, a sequence length of 10 strikes an optimal balance between speed and success. The *Suc.* performance is improved with the increase in sequence length. In contrast, the speed slows down correspondingly. The dense supervision aids the MLLM in sustaining long-term prediction capabilities, which implies that PADriver tends to be more cautious rather than aggressive with the short length.

Table 4: Ablation study on the effectiveness of the framework input, which includes the image and history states. For states, we compare the actions, coordinates, and speed elements of the ego-vehicle.

| Exp | Image | Action | Coordinates | Speed | Dis. | Spe. | Saf. | Kep. | Den. | Suc. |
|---|---|---|---|---|---|---|---|---|---|---|
| 0 | | ✓ | ✓ | ✓ | - | - | - | - | - | 0 |
| 1 | ✓ | | | | 526 | 63.11 | 0.59 | 0.94 | 0.86 | 3 |
| 2 | ✓ | ✓ | | | - | - | - | - | - | 0 |
| 3 | ✓ | | ✓ | | 550 | 65.98 | 0.68 | **0.95** | 0.89 | **23** |
| 4 | ✓ | | | ✓ | 684 | 82.08 | 0.71 | 0.85 | 0.99 | 9 |
| 5 | ✓ | | ✓ | ✓ | **723** | **86.83** | 0.67 | 0.76 | **1.01** | 17 |
| 6 | ✓ | ✓ | ✓ | ✓ | 630 | 75.49 | **0.72** | 0.70 | 1.00 | 6 |

**Scenes Description and Danger level:** We also evaluate the impact of scene descriptions and danger level on the overall performance in Table 3b. Our findings indicate that adding these elements alone can bring incremental improvements. Integrating both of them significantly enhances the efficacy of PADriver. This suggests that the chain of thought (CoT) mechanism plays a crucial role in leading to substantial performance. To further explore the distribution of danger level across three different modes, we estimate the average danger level associated with the final action in Table 3c.

The experiment reveals that the average danger level in the fast mode is significantly higher than in other modes. This finding shows that PADriver tends to opt for more aggressive actions under the fast mode, demonstrating its potential for dynamic decision-making in different modes.

**Image and Ego State:** Since the image is not that necessary for previous works (Wen et al., 2023; Fu et al., 2023) with Highway-Env and our PADriver is a hybrid (data-driven and knowledge-driven) framework, we first assess the necessity of BEV images for our method. We conduct a comparison without and with the input image, as shown in comparison (Exp.0 vs. Exp.6) of Table 4. It indicates that our model works based on the understanding of the input image.

Subsequently, we further explore the components of ego states, including the action, coordinates, and speed. Our analysis (Exp.1 vs. Exp.2) reveals that historical actions significantly affect performance. Notably, there is not even one successful completion for Exp.2. We find that introducing the historical actions brings the action shortcut, enabling the MLLM to replicate previous behaviors. We hypothesise the main reason is that the keep operation dominates the distribution of the whole dataset. As shown in Exp.(3-4), without historical actions, evaluation metrics are significantly improved when adding coordinates and speed information separately. Only adding the coordinates (Exp.3) achieves the highest *Suc.*, while driving with lower speed. When combining speed and coordinated as the ego state, *Dis.* and *Spe.* are both improved but *Suc.* drops to some extent.

## 5 CONCLUSION

In this paper, we introduce PADriver, a closed-loop framework for personalized autonomous driving that leverages a Multi-modal Large Language Models (MLLMs). By processing streaming frames, ego states, and personalized textual prompts, PADriver effectively carries out scene understanding, danger level estimation, and action decision-making in an autoregressive manner. Furthermore, a closed-loop benchmark PAD-Highway is established, which uses the Highway-Env simulator to thoroughly assess the planning performance of our approach. Our experimental results on this benchmark demonstrate that PADriver effectively achieves personalized driving with three modes: fast, normal, and slow, while achieving great driving performance across various evaluation metrics.

**Limitation and Future Work:** PADriver is currently based on Highway-Env scenes. We plan to extend it to other simulators, such as WayMax (Gulino et al., 2023) and CARLA (Dosovitskiy et al., 2017), or practical autonomous driving scenarios. Additionally, PADriver is based on the observation of Bird's Eye View (BEV) scenes, which can be extended to surround-view images or point clouds.

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

# A APPENDIX

We present more details about our method here due to the text limitation.

Table 5: (a). Comparative analysis of model transferability performance across diverse environmental conditions for environments featuring from 4 to 6 lanes and density. The 6* with the destiny of 3 compare with others of 2. (b). The illustration of the different number of lanes with different density.

| Lanes | Dis. | Spe. | Saf. | Kep. | Den. | Suc. |
|-------|------|------|------|------|------|------|
| 4 | 723 | 86.83 | 0.67 | 0.76 | 1.01 | 17 |
| 5 | 760 | 91.04 | 0.66 | 0.83 | 0.90 | 19 |
| 6 | 747 | 89.71 | 0.75 | 0.92 | 0.75 | 15 |
| 6* | 724 | 87.08 | 0.46 | 0.78 | 0.98 | 7 |

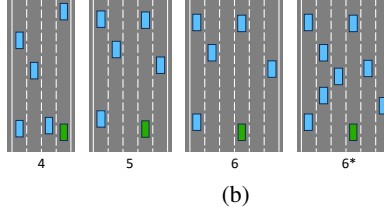

(a)                                                    (b)

## A.1 ABLATION OF TRANSFERABILITY

We also evaluate the transferability of our PADriver across various environments. We mainly focus on two variables: lane count and vehicle density. As shown in Table 5, the results show that the *Success Number (Suc.)* increases when the number of lanes increases (4→5). This may owing to the lower vehicle density while the scene change is minor. Conversely, when the number of lanes is directly increased from 4 to 6, a small performance decline was observed. When vehicle density was proportionally increased to match the expansion from 4 to 6 lanes (4→6*), the *Suc.* metric significantly dropped, mirroring the scene's similarity associated with the 4-lane.

Table 6: Ablation study on the data size. We evaluated the function of the pretraining and SFT stages, taking into account the data size of the SFT stage with half of SFT data.

| Exp | Pertaining | SFT | Dis. | Spe. | Saf. | Kep. | Den. | Suc. |
|-----|-----------|-----|------|------|------|------|------|------|
| 0 | | ✓ | - | - | - | - | - | 0 |
| 1 | ✓ | | 751 | 90.10 | 0.62 | 0.82 | 0.76 | 9 |
| 2 | ✓ | $\frac{1}{2}$ | 721 | 86.60 | 0.65 | 0.78 | 0.80 | 15 |
| 3 | ✓ | ✓ | 723 | 86.83 | 0.67 | 0.76 | 1.01 | 17 |

## A.2 ABLATION OF THE DATA SIZE

As shown in Table 6, we assess the effectiveness of pretraining and the SFT strategy. Initially, the pretraining stage is omitted from the training, and SFT is directly applied to assess the effectiveness of the pretraining stage. Subsequently, we examine the performance of the PADriver based solely on the pretraining stage. It is important to note that in the pretraining stage, the model's input does not include personalized prompts. However, the PADriver with only pretraining retains transferability. Lastly, we utilize half of the fine-tuning data to evaluate the performance of the framework, highlighting that even in the SFT stage, data size is important.

## A.3 DANGER LEVEL ESTIMATION

Danger levels are explicitly modeled based on a comprehensive analysis of the Bird's Eye View (BEV) scenes. As shown in Algorithm 1, it incorporates some factors such as the anticipated reaction time, the velocity of the ego vehicle, and the surrounding environmental conditions—namely, the proximity to and velocity of nearby vehicles.

---

**Algorithm 1:** Generation of danger level

---

**Input:** Ego vehicle's state $S_e = \{(x_e, y_e), v_e\}$; Other $n$ vehicles' states
       $S_o = \{(x_i, y_i), v_i | i \in [0, n]\}$, $(x, y)$ and $v$ are the coordinates and speed of vehicles,
       respectively.
**Output:** Danger levels for five actions $D = \{d_l, d_k, d_r, d_f, d_s\}$, where $d_l$, $d_k$, $d_r$, $d_f$, and $d_s$
       correspond to the danger level of `<left>`, `<keep>`, `<right>`, `<faster>`, and
       `<slower>` actions, respectively.

**1** `CalculateDanger`: Calculate the danger level based on speed and distance.
**2** `InSameLane`: Determine whether the car is in the same lane as the ego car.
**3** `BesideEgo`: Determine whether the car is beside the ego car.
**4** Initialize the danger levels $D = \{$`<0>`, `<0>`, `<0>`, `<0>`, `<0>`$\}$;
**5 for** $S_i \in S_o$ **do**
**6**      Calculate $\Delta x = x_e - x_i$, $\Delta y = y_e - y_i$, $\Delta v = v_e - v_i$;
**7**      **if** `InSameLane` $(\Delta_y)$ **then**
**8**          $\{d_k, d_f, d_s\} = \max(\{d_k, d_f, d_s\}$, `CalculateDanger` $(\Delta x, \Delta v))$;
**9**      **else**
**10**          **if** `BesideEgo` $(\Delta x, \Delta y)$ **then**
**11**              $\{d_l, d_r\} = \max(\{d_l, d_r\}$, `<NOT>`$)$;
**12**          **else**
**13**              $\{d_l, d_r\} = \max(\{d_l, d_r\}$, `CalculateDanger` $(\Delta x, \Delta v))$;
**14**          **end**
**15**      **end**
**16 end**

---

### A.4 THE ILLUSTRATION

We demonstrate the performance of the PADriver with three modes and the Dilu (Wen et al., 2023) framework. The video demonstrating this performance is in the attached supplement files **seed0.mp4** and **seed20.mp4**, corresponding to seeds 0 and 20, respectively.

### A.5 THE DETAILED INSTRUCTION OF THE INPUT AND OUTPUT

As shown in Figure 4, the instructions for the PADriver are as follows. The PADriver framework takes textual prompts and the BEV image as input. The textual prompts include system prompts, personalized prompts, and ego state queues. After tokenizing them into textual and vision tokens, the LLM then autoregressively generates scene descriptions, assesses the danger level for each potential action, and makes the final action decision. The framework generates the final action and the subsequent nine actions for long-term prediction. The final action is implicitly affected by both the danger level estimation and the scene description. The system prompt allows permutations with different sentences, as shown in Tables 7, 8, and 9. We also provide different mode instructions in Table 10.

**In Frame$_t$ :**

**System Prompt:** You, the 'ego' car, are now driving a car on a road. The green box is 'ego' car, the blue box is other vehicles, the gray box is the historical trajectory of the car, the gray background is the ground, the white solid line is the edge of the road, and the white dotted line is the lane line.
There are several rules you need to follow when you drive on a highway:

**Traffic Rule:**
1. Try to keep a safe distance from the car in front of you. If there is no safe distance, double-check the safety of the target lane. If there is no car beside you, you can take the action of <left> or <right>, which means change lane left or change lane right.\n
2. If you find nothing ahead, you can take the action of <faster>, which means accelerate or faster, or take the action of <keep>, which means keep_speed or idle, to maintain your current state.\n
3. If you find yourself behind slower traffic and deceleration seems like the only option, you can take the action of <slower>, which means decelerate or slower. Please make decision for the 'ego' car. Let's think step by step, first estimate the risk of each action, then analyze the possible actions, and finally output your action. The comparative table of output decision action is: {'<left>': 'change lane left', '<keep>': 'keep_speed or idle', '<right>': 'change lane right', '<faster>': 'accelerate or faster', '<slower>': 'decelerate or slower'}.

**Personal instruction:** Please prioritize safety and comfort as much as possible, which means avoiding taking action <faster>, corresponding to accelerate or faster, and avoiding frequent lane changes.

**Ego state Prompt:** The state 1 is: the speed is 25 m/s and the position is (308, 8). The state 2 is: the speed is 25 m/s and the position is (310, 8). The state 3 is: the speed is 25 m/s and the position is (313, 8). The state NOW are: the speed is 25 m/s and the position is (315, 8).
**Image:** The frame NOW is

The scenes NOW are

**Answer:**
**Scenes description**: It's safe on the left. The right sector is free from danger. The vehicle ahead is at a minimal distance.

**Danger level**: Therefore, the danger scores are *<keep>* is *<7>*; *<left>* is *<1>*; *<right>* is *<5>*; *<faster>* is *<9>*; *<slower>* is *<4>*;

**The final output:** The next action is **<right>**, which means change lane right. The future action 1 is <keep> , which means keep_speed or idle. The future action 2 is <keep> , which means keep_speed or idle. The future action 3 is <keep> , which means keep_speed or idle. The future action 4 is <keep> , which means keep_speed or idle. The future action 5 is <keep> , which means keep_speed or idle. The future action 6 is <keep> , which means keep_speed or idle. The future action 7 is <keep> , which means keep_speed or idle. The future action 8 is <keep> , which means keep_speed or idle. The future action 9 is <faster> , which means accelerate or faster.

Figure 4: The illustration of the full input and output of the PADriver.

Table 7: Examples of system prompt for the basic environment descriptions of the ego car.

1. "You, the 'ego' car, are now driving a car on a road."

2. "You, embodying the 'ego' vehicle, are currently maneuvering a sleek sedan down a bustling city street."

3. "As the 'ego' car, you find yourself smoothly cruising along a winding country road, surrounded by nature."

4. "You, operating as the 'ego' car, are navigating through a densely packed highway with precision and care."

5. "In the role of the 'ego' car, you are gently steering a family car down a quiet, suburban neighborhood street."

6. "You, the 'ego' car, are currently threading through traffic in an urban setting, with skyscrapers towering above."

7. "As the 'ego' car, you're driving a convertible along a picturesque coastal road, with the ocean breeze in your hair."

8. "You, in the capacity of the 'ego' car, are maneuvering a compact vehicle through a maze of narrow, cobbled streets in an old town."

9. "You, the 'ego' car, are now gliding along a deserted road that cuts through a vast, serene desert landscape."

10. "As the 'ego' car, you find yourself at the helm of a rugged SUV, traversing a rough, mountainous terrain."

11. "In the role of the 'ego' car, you're piloting a luxury car down a glamorous, tree-lined boulevard in a posh neighborhood."

Table 8: Examples of system prompt for the BEV scenes description.

1. "The green box is 'ego' car, the blue box is other vehicles, the gray box is the historical trajectory of the car, the gray background is the ground, the white solid line is the edge of the road, and the white dotted line is the lane line."

2. "The vivid green box represents the 'ego' car, while the azure blue box delineates other vehicles; the slate gray box traces the car's past path, set against a monochrome gray backdrop signifying the ground, bordered by the stark white lines marking the road's boundaries and the dashed lines indicating the lanes."

3. "A bright green box symbolizes the 'ego' car, surrounded by sky blue boxes for nearby vehicles; a muted gray box outlines the vehicle's previous route, all on a neutral gray canvas representing the ground, with clean white lines framing the road's edges and speckled lines partitioning the lanes."

4. "In this schematic, the 'ego' car is a lime green box, other vehicles are marked by cerulean blue boxes, and the car's history is a charcoal gray box, all laid out on a gray ground, with the road's periphery and lanes defined by unblemished white lines and intermittent dashes, respectively."

5. "Here, the 'ego' car is encapsulated within an emerald green box, contrasted by cobalt blue boxes for other vehicles and a smoky gray box mapping the car's trajectory, all against a stone gray ground, flanked by pure white lines demarcating the road's edge and the segmented lines allocating the lanes."

6. "The scene features the 'ego' car as a forest green box, other vehicles as navy blue boxes, and the car's historical path as a steel gray box, all against a matte gray surface symbolizing the ground, with the road's outline and lane separations clearly defined by continuous and dotted white lines, respectively."

7. "In this visual, the 'ego' car is indicated by a mint green box, with sapphire blue boxes for other vehicles and a dove gray box for the car's past path, all positioned on a silvery gray ground, bordered by the pristine white lines of the road's extremities and the punctuated lines demarcating the lanes."

8. "This representation shows the 'ego' car as a jade green box, other vehicles as royal blue boxes, and the car's historical route as an ash gray box, all against a pewter gray ground, with the road's periphery and lane divisions etched in immaculate white solid and dashed lines."

9. "In this depiction, the 'ego' car is a shamrock green box, juxtaposed with indigo blue boxes for other vehicles and a shadow gray box tracing the car's former trajectory, all set upon a slate gray ground, edged by the road's crisp white border and the punctuated lines that segment the lanes."

10. "The layout presents the 'ego' car as an olive green box, with other vehicles as lapis blue boxes and the car's historical pathway as a fog gray box, all over a charcoal gray ground, with the road's margins and lane separations crisply delineated by solid and dotted white lines."

11. "Here, the 'ego' car is a hunter green box, other vehicles are in teal blue boxes, and the car's previous movements are in a graphite gray box, all against a flint gray ground, with the white lines cleanly defining the road's edge and the dash-lined lanes."

Table 9: Examples of system prompt for the other appropriate notices and instructions.

1. "Please first describe the scene, then estimate the risk of each action."

2. "Initially, provide a depiction of the scene, followed by an assessment of the risk associated with each possible action."

3. "Begin by detailing the scene, then proceed to evaluate the potential risk entailed by each action."

4. "First, portray the setting of the scenario, and subsequently, analyze the risk level of every action."

5. "Start with a description of the scene, then move on to ascertain the risk involved in each specific action."

6. "Initially, paint a picture of the scene, then methodically gauge the risk each action carries."

7. "Commence by depicting the scene, and then proceed to estimate the risk factor for each action."

8. "First, lay out the scene in detail, then evaluate the risk associated with each action taken."

9. "Begin by giving a visual account of the scene, followed by a risk estimation for each action."

10. "Start with a narrative of the scene, and then progress to determine the risk level of each possible action."

11. "Lead with a comprehensive description of the scene, then assess the risk implicated by each action."

Table 10: Examples of personalized prompts.

- FAST_INSTRUCTION: "Please maintain speed as much as possible, which means avoiding taking action <slower>, which corresponds to decelerate or slower."

- NORMAL_INSTRUCTION: "Please prioritize safety and comfort as much as possible, which means avoiding frequent lane changes."

- SLOW_INSTRUCTION: "Please keep safety by avoiding taking action <faster>, corresponding to accelerate or faster, and minimizing lane changes. "

