# OpenReview forum: "PADriver: Towards Personalized Autonomous Driving"
_ICLR.cc/2025/Conference — ICLR 2025 Conference Withdrawn Submission_

### Official Review · Reviewer_PPug · 2024-10-27

**Soundness:** 2
**Presentation:** 2
**Contribution:** 2
**Rating:** 3
**Confidence:** 4

**Summary:**

This paper presents PADriver, an MLLM that takes a rasterized BEV image of traffic scenario as well as a personalized prompt as the inputs, and then outputs the driving actions that follow the driving behavior described in the personalized prompt. The paper also presents a dataset called PAD-Highway, which consist of partly rule-generated and partly human-based driving logs in HighwayEnv scenario.

**Strengths:**

1. The paper is clearly motivated.

This paper has a very clear motivation: enabling personalized autonomous driving with MLLM. This is an interesting direction and can potentially lead to interesting technical contributions or datasets.

2. The paper provides human-labeled logs in the proposed PAD-Highway dataset.

In the PAD-Highway dataset, this work presents 25 hours of human-based driving logs and scores for driving experience in the HighwayEnv environment. I believe this collection of human-labeled data will be useful for future works.

**Weaknesses:**

1. The necessity of making PADriver a MLLM for HighwayEnv is unclear.

The most important contribution of this paper is the proposed PADriver, which is an MLLM trained for HighwayEnv. As mentioned in Line044, the main reason is that only providing text description is not sufficient to capture enough scene information. Therefore, they use vision inputs with BEV rasterizations.

However, in fact, the BEV image from HighwayEnv does not provide any additional information than all the agent's locations in the scenario, which can be easily extracted as float numbers and provided as either text input or vectorized low-dimensional features. Therefore, it is unclear to me why is using vision input necessary and what benefits it provides for HighwayEnv. It seems to me using text inputs or simply agent-location vectors will be sufficient.

In short, I agree that vision input might be important for more realistic environments like CARLA. However, as this work only conducts experiments on HighwayEnv, the purpose of using MLLM is unclear.

2. The necessity to train the MLLM PADriver instead of directly prompting SOTA MLLMs is unclear.

Related to the last part, PADriver is trained with a large amount of driving data generated by rule-based policies as well as human annotations. However, because the BEV rasterization in HighwayEnv is a very easy and intuitive representation of the scenario, it might be very likely that current MLLMs (e.g., GPT4o) might be already able to solve the PAD problem with relatively good performance.

DILU has already proved that pure-text GPTs are already able to do well in HighwayEnv with text input, I believe with proper prompting and input formatting, it is very straightforward to modify DILU's prompts and use GPT4o to take in the image input and output personalized driving behaviors. To show the necessity of training an MLLM to solve this issue, I believe including GPT4o's performance is important.

3. The use of personalized driving prompts is very limited.

Although the paper has an emphasis on "personalized driving prompts", the kind of personalization is only limited to "slow, medium, and fast" and do not include very detailed personalized prompting like "always try to drive on the leftmost side", "try to keep a distance of 10m from the car in front", "do not exceed acceleration of XX level".

The reason why the detailed personalized prompts are expected is that the PADriver is an MLLM, it is very natural for it to take in free-form and high informative language descriptions of the personalized needs. The current support for 3 levels of speeding can be simply input to the model with a one-hot vector with 3 dimensions, without any text involved.

As personalized driving is the core focus of this paper, I think this limitation becomes an important drawback of this work.

**Questions:**

1. Table 3: is Table 3 having a fair comparison? Need to highlight the differences.

In Table 3 several compared methods are included, some of them (e.g., DILU) do not take the same input format, some of them are not trained or finetuned on the same dataset as PADriver. To compose a fair comparison and understand why PADriver is doing well, it is necessary to highlight these important differences.

2. Line315: What's the performance of the rule-based policy in the proposed benchmark?

A large portion of the driving data in the PAD-Highway dataset is generated by the rule-based method, and PADriver is trained on this set of generated data. It becomes very interesting how the rule-based policy perform on the benchmark in the paper; after all PADriving is imitating this rule-based policy.

If this rule-based policy can perform well on the benchmark, the necessity to train PADriver becomes questionable.

---

### Official Review · Reviewer_FWuf · 2024-10-29

**Soundness:** 2
**Presentation:** 2
**Contribution:** 2
**Rating:** 3
**Confidence:** 4

**Summary:**

This paper presents PADriver, a framework for personalized autonomous driving (PAD) using Multi-modal Large Language Models (MLLMs). PADriver integrates real-time frames and personalized prompts for scene understanding, danger level estimation, and action decisions. The authors also create PAD-Highway, a benchmark with 250 hours of annotated data for evaluating PAD systems.

**Strengths:**

1. PADriver introduces a novel closed-loop system for personalized autonomous driving, utilizing Multi-modal Large Language Models for comprehensive scene understanding and decision-making.

2. The introduction of a "danger level" metric to evaluate the risk of potential actions enhances the accuracy and safety of the driving decisions, setting PADriver apart from existing methods.

3. The PAD-Highway, a benchmark with 250 hours of high-quality annotated data, provides a robust platform for evaluating and advancing personalized autonomous driving technologies.

4. The experiment results of different driving styles are very interesting.

**Weaknesses:**

1. The current implementation of three driving styles (slow, fast, normal) is limited to speed variations. Expanding the range of driving styles could better reflect the "Personalized" aspect emphasized in the title.

2. The experiments are solely based on Vicuna-7B-1.5. It would be beneficial to test other LLM weights, such as LLaMA3, LLaVA, and Vicuna-13B-1.5, to evaluate performance across different models.

3. The driving personalities are currently represented through statistics. Providing more visualizations or trajectory plots of different driving styles could offer a more intuitive understanding of the results.

4. The need for images in the framework is not fully justified. An experiment removing images from the inputs would clarify if textual descriptions alone can sufficiently describe the scenarios.

5. The related work on end-to-end autonomous driving methods is insufficient. Many recent imitation learning based works are missing, such as Interfuser[1], ReasonNet[2], DriveAdapter[3], etc,.

6. Line 191 has a format problem. The format of the subsubsection is inconsistent in the paper.

[1]: InterFuser: Safety-Enhanced Autonomous Driving Using Interpretable Sensor Fusion Transformer

[2]: A New Paradigm for End-to-end Autonomous Driving to Alleviate Causal Confusion

[3]: ReasonNet: End-to-End Driving with Temporal and Global Reasoning

**Questions:**

N/A

---

### Official Review · Reviewer_BGji · 2024-11-03

**Soundness:** 2
**Presentation:** 3
**Contribution:** 2
**Rating:** 5
**Confidence:** 4

**Summary:**

This paper proposes an interesting topic and novel framework called PADriver for personalalized autonomous driving. Different from conventional autonomous system, the proposed model is mainly based on langugae driven simulated environment, which would be more simple than traditional autonomous driving task.

**Strengths:**

1. The personalized autonomous driving topic is interesting and motivating.
2. The reported results demonstrate better performance than existing methods on the proposed benchmark.

**Weaknesses:**

1. The visual environment in this simulation is very simple, which means the proposed method would not be practical in real-world applications. Additionally, it seems this paper mainly addresses motion planning. Based on these, the visual environment should be more complicated if the visual encoder is used, such as changing the shape and color of the simulated "cars". Otherwise, the visual encoder should not be used and only use langugae modality for motion planning.

2. The information between different modalities should not include each other. As shown in Figure 2, it doesn't make sense that the system prompt directly points out color information in the visual images.

**Questions:**

In a real autonomous driving system, perception is very important for motion planning. It usually has many FP and FN samples in the detected results, this paper should consider this.

---

### Official Review · Reviewer_u3q3 · 2024-11-04

**Soundness:** 3
**Presentation:** 3
**Contribution:** 3
**Rating:** 5
**Confidence:** 4

**Summary:**

This paper proposes an auto-driving framework based on the Multi-modal LLM by introducing the personal prompt for personal drive mode adaptation to make action decisions with consideration of scene understanding and danger level as an E2E manner. To test the method, a new benchmark PAD-HIGHWAY is proposed based on the high-way simulator for evaluating different closed-loop methods and show the proposal advantages on the top of the benchmark.

**Strengths:**

- The motivation of this paper is regarding the discussion about the personalized auto drive system designed to provide a general auto-drive mode preference adopted to the end user and changeable from the personal prompt which is a great application direction.

- The design of the drive mode is defined as  slow, fast and normal as the general preference is well balanced, and the framework involves the scene description, danger level to do the action decision is technical sounds.

- The proposed dataset is relatively large scale and annotated with action, speed, coordinates which are useful to test if the action decision is correct or not.

- The evaluation focuses on testing 7 specific indexes regarding drive experience and is novel and interesting, which is not only a final output evaluation but also evaluating the behavior to achieve it during the drive with more human-centric consideration

**Weaknesses:**

- The previous methods focus on individual modeling of personal auto-drive could provide a different value as data-driven personal design of the driving, which is not a drawback. Author should put more focus on the method comparison, as general preference personalization and individual personalization are two different directions of the applications.

- Prompt setup is the key for the framework, and it lacks of the detail contents of it, e.g., traffic rule, as a general pre-set prompt, what the principle to design it, it seems if the rule is define by rule-base or limited items, does it enough to cover the different scenarios in the real world.

- Slow, normal, fast how it is utilized in the pipeline is not clear, it seems the mode is defined in the personal prompts, but the example prompt can not find the clue of the definition of it.

- The train process is not clear, which loss and which part is the train target is not mentioned. It suggests using the math symbol and equations to describe the data and train process.

- Proposed dataset is unbalanced for use, and the train on the top of it may cause the result to be preferred to give the action which is the major in the dataset to acquire high scores.

- The most concerns are, scene understanding and danger level estimation as the base information for the action decision are not evaluated to show how accurate it is, as the two important parts of the model.

- Table 3, ablation study should keep the Mode as the same; Comfort metrics need to give clear definition;

**Questions:**

- If the degree of danger is increased with the increasing level number, in the given example, why <left> is 1 indicates the danger level is low.

- How to annotate the scene description.

- What is the reason behind the difference among the slow, normal and fast.

- It recommended giving a figure about how the proposed dataset and its annotation looks like.

The problem setting and method with a new dataset are impressive for the efforts, I would like to see the responses from the author to  give the final decision.

---

### Note · Authors · 2024-11-13

I have read and agree with the venue's withdrawal policy on behalf of myself and my co-authors.